# Human Induced Pluripotent Spheroids’ Growth Is Driven by Viscoelastic Properties and Macrostructure of 3D Hydrogel Environment

**DOI:** 10.3390/bioengineering10121418

**Published:** 2023-12-13

**Authors:** Lucas Lemarié, Tanushri Dargar, Isabelle Grosjean, Vincent Gache, Edwin J. Courtial, Jérôme Sohier

**Affiliations:** 1SEGULA Technologies, 69100 Villeurbanne, France; lucas.lemarie@segula.fr; 23d.FAB, CNRS UMR 5246, ICBMS (Institute of Molecular and Supramolecular Chemistry and Biochemistry), Université Lyon 1, 69622 Villeurbanne, France; edwin.courtial@univ-lyon1.fr; 3CNRS UMR 5305, LBTI (Tissue Biology and Therapeutic Engineering Laboratory), 69007 Lyon, France; 4CNRS UMR5261, INSERM U1315, INMG-PNMG (NeuroMyoGene Institute, Physiopathology and Genetics of the Neuron and the Muscle), Université Lyon 1, 69008 Lyon, France; tanushri.dargar@univ-lyon1.fr (T.D.); isabelle.grosjean@univ-lyon1.fr (I.G.); vincent.gache@univ-lyon1.fr (V.G.)

**Keywords:** human iPSCs, spheroids, growth, stem cells, hydrogel, viscoelastic properties, porosity

## Abstract

Stem cells, particularly human iPSCs, constitute a powerful tool for tissue engineering, notably through spheroid and organoid models. While the sensitivity of stem cells to the viscoelastic properties of their direct microenvironment is well-described, stem cell differentiation still relies on biochemical factors. Our aim is to investigate the role of the viscoelastic properties of hiPSC spheroids’ direct environment on their fate. To ensure that cell growth is driven only by mechanical interaction, bioprintable alginate–gelatin hydrogels with significantly different viscoelastic properties were utilized in differentiation factor-free culture medium. Alginate–gelatin hydrogels of varying concentrations were developed to provide 3D environments of significantly different mechanical properties, ranging from 1 to 100 kPa, while allowing printability. hiPSC spheroids from two different cell lines were prepared by aggregation (⌀ = 100 µm, n > 1 × 10^4^), included and cultured in the different hydrogels for 14 days. While spheroids within dense hydrogels exhibited limited growth, irrespective of formulation, porous hydrogels prepared with a liquid–liquid emulsion method displayed significant variations of spheroid morphology and growth as a function of hydrogel mechanical properties. Transversal culture (adjacent spheroids-laden alginate–gelatin hydrogels) clearly confirmed the separate effect of each hydrogel environment on hiPSC spheroid behavior. This study is the first to demonstrate that a mechanically modulated microenvironment induces diverse hiPSC spheroid behavior without the influence of other factors. It allows one to envision the combination of multiple formulations to create a complex object, where the fate of hiPSCs will be independently controlled by their direct microenvironment.

## 1. Introduction

Human induced pluripotent stem cells (hiPSCs) are a formidable tool for applications in tissue engineering, microchip development and cancer diagnosis [1,2,3,4,5]. Their ease of production through reprogramming methods overcomes the conventional limitations associated with classical stem cell applications. This versatility enables enhanced disease modeling, tissue engineering and autologous cell-based therapy [6,7,8]. Consequently, hiPCSs have been successfully employed to generate artificial tissues, such as the heart, retina, gut, skin and more [9,10,11,12,13,14], with the goal of scaling up and complexifying these models using techniques like 3D bioprinting [15,16,17] or electrospinning [18].

Despite the extensive exploration of hiPSCs for various applications, there are still significant challenges associated with their use. For instance, existing models and their associated environments are not optimal for maintaining stemness. Planar cell culture on rigid plastic surfaces is gradually being replaced by methods that incorporate biomaterials and 3D environments [19,20]. Similarly, the use of isolated cells is now being challenged by the introduction of spheroids, which has been demonstrated to promote cell differentiation or the secretion of trophic factors from stem cells [21,22]. While it is clear that cell differentiation is influenced by biochemical, biophysical and biomechanical factors, standard methods continue to rely on differentiation factors to biochemically guide cells towards specific lineages. This suggests that certain fundamental mechanisms remain elusive.

Additionally, the use of easily accessible, cost-effective and reproducible biomaterials is a challenge facing the scientific community. The widespread use of Matrigel raises concerns about reproducibility and batch-to-batch variability, in addition to supply limitations, due to its origin (basal membrane extracted from a mouse sarcoma) and production method. Furthermore, Matrigel does not provide the means to modulate the viscoelastic properties of the cellular environment, and there is currently no material that offers a solid alternative with variable mechanical properties [23,24]. This is a critical consideration for the potential standardization of hiPSCs culture, given that cells, especially stem cells, are sensitive to their mechanical environment [25,26,27].

The macrostructure of the environment is also a crucial factor in cell proliferation. Studies have shown that the presence of porosity enhances cell growth and spreading by maximizing cell–material interaction [28,29,30] while facilitating optimal diffusion of the culture medium without the need for microenvironment remodeling or degradation [31,32]. For stem cell models, the presence of porosity appears particularly relevant for inducing stemness niches while allowing for cell differentiation [33].

A wide array of natural and synthetic hydrogels with the necessary biochemical and mechanical properties for replicating the cellular environment, particularly for stem cells, is available [20,34,35,36,37,38,39,40]. Alginate and gelatin hydrogels, in particular, are widely used and described as a substantial compromise between high biocompatibility, affordability and tunability. In this context, the viscoelastic properties of alginate–gelatin hydrogels can be readily adjusted by varying their concentration [41,42], and microporosity can be easily incorporated within the hydrogels [43]. This justifies the widespread use of alginate–gelatin in 3D bioprinting [44,45,46] and, more broadly, in tissue engineering [47].

Hence, to better elucidate the role of the 3D environment properties on hiPSCs spheroid behavior, porous hydrogels made from alginate/gelatin with customizable mechanical properties were developed, alongside a spheroid induction method. Subsequently, the effect of viscoelastic properties and macrostructure on hiPSCs spheroid growth in a porous environment was then investigated, without the use of growth factors.

## 2. Materials and Methods

### 2.1. (Porous) Alginate–Gelatin Hydrogels Preparation

Alginate–Gelatin (AG) hydrogels were prepared by dissolving sodium alginate (120–190 kDa, 39% guluronic acid, 180947-100G, Sigma, St. Louis, MO, USA) and gelatin (40–100 kDa, type B, G9382, Sigma, USA) in DPBS (Dulbecco’s Phosphate Buffered Saline, Thermo Scientific, Waltham, MA, USA). Three formulations were created with a 1:2 ratio of alginate to gelatin: AG-1X (1% *w*/*v* alginate, 2% *w*/*v* gelatin), AG-3X (3% *w*/*v* alginate, 6% *w*/*v* gelatin), and AG-5X (5% *w*/*v* alginate, 10% *w*/*v* gelatin). To prepare each formulation, the respective amounts of sodium alginate and gelatin powders were weighed and placed in a volumetric flask. They were then dissolved in DPBS at room temperature to reach a final volume of 100 mL. The AG blends were maintained at 90 °C for 2 h to ensure complete dissolution of the components and were then subsequently pasteurized. After sterilization, the AG hydrogels were sealed and stored at 4 °C until further experiments.

To introduce porosity in the hydrogels, a liquid–liquid emulsion method was employed [28]. Briefly, a sterile solution of PEG (PolyEthylene Glycol 20 kDa, Sigma, USA) was prepared by dissolving 0.5 g in 1 mL of distilled water (50%, *w*/*v*) under mechanical stirring for 30 min. This solution was subsequently added to the AG-nX formulations (n = 1, 3, 5) at a volume ratio of 10%. A 500 µL emulsion was produced by 10 suction-reflow movements of 50 µL in a 1.5 mL eppendorf tube, using a 100 µL positive displacement pipet (MicromanTM, Thermofisher, Franklin, MA, USA) at 15 °C for the AG-1X, 25 °C for the AG-3X and 37 °C for the AG-5X. The resulting porous hydrogel are denoted with an asterisk (AG-nX*) to distinguish them from the dense hydrogel (AG-nX).

For experimentations and characterization, each hydrogel was manually extruded into the well of a 24-well plate. After curing (10 min at 4 °C to allow gelation), a cross-linking solution containing 1% *w*/*v* CaCl_2_ (Sigma, USA) and 1% *w*/*v* transglutaminase (Ajinomoto, Chūō, Tokyo, Japan) was applied to the discs for 10 min at 37 °C.

### 2.2. Mechanical and Rheological Characterization

Viscoelastic characterization of the hydrogels was performed at 37 °C using a Discovery Hybrid Rheometer (DHR) 2 (TA instruments, New Castle, DE, USA). To determine elastic properties, Young’s modulus was derived through Dynamic Mechanical Analysis (DMA) performed on dense hydrogel discs with a diameter of 40 mm and a thickness of 2 mm.

A second-order Generalized Maxwell (Appendix A) model was used to determine the storage and loss moduli (E′ and E″, respectively, Appendix A), which allowed for the calculation of Young’s modulus. The comparison of experimental and calculated data is shown in Appendix A, and the various model parameters (β, τ and E) are shown in Appendix A.

For the investigation of the viscous property, a relaxation test was used [48] to determine τ_1/2_ and quantify long time relaxation (>15 min). A constant compression stress was applied using a 40 mm-diameter geometry on the hydrogels prepared in the same manner after confirming the linear viscoelastic region (between 20 and 200 Pa) over 1500 s.

### 2.3. Structural Characterisation

Dense hydrogel discs, with a diameter of 20 mm and thickness of 1 mm were prepared and frozen in liquid nitrogen. Following this, the discs were placed between two smooth metal jaws and broken by mechanical impact to reveal their microstructure without introducing any cutting artifacts. Samples were placed on their edges and image acquisition was carried out using a scanning electron microscope (SEM) (TM4000, Hitachi, Tokyo, Japan) at 10 kV in backscattered electron (BSE) mode at 300× magnification.

### 2.4. hiPSCs Culture

Two human induced pluripotent stem (hiPSCs) cell lines were employed in this study. The AG08C5 line is officially registered in the European hPSCreg as PGNMi001-A (https://hpscreg.eu/cell-line/PGNMi001-A, accessed on 12 November 2023). We also acknowledge that all human cells utilized at the iPS-PGNM platform are duly declared to the French Ministry of Health (CODECOH DC-2022-5055). The SCTi003-A is a commercial cell line, female, and derived from blood cells (STEMCELL, Vancouver, BC, Canada).

The hiPSCs cell lines were cultured in mTeSR+ medium (STEMCELL, Canada) on plates coated with Vitronectin XFTM (STEMCELL, Canada) until they reached 80–90% confluence. The medium was changed every 2 days during the amplification process. Subsequently, the medium was aspirated, and the cells were washed with 2.5 mL DPBS at room temperature. Then, the cells were detached using 2 mL of TrypLE^TM^ (Thermofisher, USA) for 5 min at 37 °C. The entire solution was collected, and the action of TrypLE^TM^ was neutralized by adding 1% foetal bovine serum (FBS, Sigma, USA). After a 4 min centrifugation at 200× *g*, the cell pellet was dissociated by flushing and used for spheroid formation.

### 2.5. Spheroids Formation, Inclusion, Maturation, and Fixation

The formation of spheroids was based on an approach adapted from a previous study [49]. A silicon mold (BLUESILTM RTV 3503, Elkem Silicones, Kristiansand, Norway) was created using a resin-printed negative mold (Acrylate-like material, e.g., Vero ClearTM, Stratasys, Rehovot, Israel). This mold was customized to fit the diameter of a well in a 6-well plate well and contained 2000 micro-wells, each with a diameter of 500 µm. After sterilizing the silicone discs by autoclaving, the discs were placed in the wells of a 6-well plate. A solution of mTeSR™ Plus (STEMCELL, Canada) supplemented with 0.1% *v*/*v* ROCK inhibitor (Y-27632, STEMCELL, Canada) containing 1.10^6^ cells was deposited onto the silicone discs.

After allowing the cells to settle for 1 h at 37 °C, a centrifugation step at 200× *g* for 5 min was carried out, and the plate was incubated at 37 °C (5% CO_2_) for 24 h. The spheroids obtained (approximately 2000 per disc) were collected by mechanical flushing and centrifugation at 200× *g* for 1 min. The delicate pellets obtained were gently aspirated and incorporated into the hydrogels by gentle suction-reflow movements until complete homogenization, for a ratio of 2000 spheroids for 1 mL of AG-nX* (where n = 1, 3, 5). Once included in the hydrogels, each sample was grown in DMEM (Dulbecco’s Modified Eagle Medium, ThermoFisher, Waltham, MA, USA) supplemented with only ITS (Insulin Transferin Selenium, Corning™, New York, NY, USA) and without differentiation factors. The medium was changed every two days until maturation was completed (Day 14). The samples were then fixed in 4% PFA for 15 min.

For transversal experiments, AG-nX* (where n = 1, 2, 3) with included spheroids were deposited in the same well through manual side-by-side extrusion. This method facilitated the fabrication of a planar structure comprising alternating fibrous layers composed of distinct hydrogel compositions. The extrusion process was carried out employing a 1 mL syringe equipped with a 3 mm long nozzle possessing an 800 µm diameter. Subsequently, the fabricated constructs underwent a series of procedures, including crosslinking, culture and fixation, which were executed following the previously outlined protocol.

### 2.6. Porosity Distribution Quantification

AG-nX (where n = 1, 3, 5) hydrogels were stained in a solution of rhodamine B at a concentration of 1 µg/mL (Fluka, Monte Carlo, Monaco) for 10 min. Stacks with a depth of 200 nm were captured at 584 nm using a confocal microscope (Zeiss 880, Oberkochen, Germany). The “Image Processing” toolbox of Matlab^®^ software (R2020b) was employed for thresholding and quantification operations. Segmentation functions (“ImAdjust and ImBinarize”) and mathematical morphology functions (“ImOpen, ImClose, bWareaopen”) were used to quantify the pore area and the proportion of porosity (total surface versus porous surface). The processed images were used to reconstruct a 3D stack using the Matlab^®^ Volume Viewer (R2020b).

### 2.7. Spheroid Growth Quantification

Images of spheroids in AG-nX (where n = 1, 3, 5) were captured using an optical microscope CKX53 (Olympus, Tokyo, Japan). The “Image Processing” toolbox of Matlab^®^ software (R2020b) was employed for thresholding and quantification procedures. These included the “ImFindCircle” function, which detects spheroids in an image and measures their diameter. Additionally, segmentation functions (“ImAdjust and ImBinarize”) and mathematical morphology functions (“ImOpen, ImClose, bWareaopen”) were used for quantifying spheroid area over time and under different experimental conditions.

### 2.8. Statistical Analysis

Statistical analyses were conducted using Graphpad Prism. For the assessment of viscoelastic properties and the hydrogel fraction, ANOVA tests were employed after a Shapiro–Wilk normality test. In the case of growth analysis, the quantified area values were subjected to log transformation, and ANOVA tests were used after a Shapiro–Wilk lognormality test. To compare pore area, the Kruskal–Wallis test was performed, and for spheroid diameter comparison, the Mann–Whitney test was utilized. Error bars represent the standard error of the mean (s.e.m.) and the quantitative data were derived from three or more independent experiments.

## 3. Results

### 3.1. Mechanical and Structural Properties of the Porous AG-nX (n = 1, 3, 5) Hydrogels

Viscoelastic measurements show that elasticity exhibited a proportional increase in accordance with the concentration of alginate/gelatin (Figure 1a). Conversely, the relaxation time exhibited a proportional decrease relative to the concentration of alginate/gelatin, as depicted in Figure 1b.

The difference in the hydrogels’ concentration was also evident in the microstructure of the porous hydrogels created using the liquid–liquid emulsion method. As depicted in Figure 2a, the microporosity of the hydrogels exhibited a significant diameter in the porous AG-1X formulation, while it was reduced in the porous AG-3X formulation and reached a minimum in the porous AG-5X formulation. It is important to note that this observation was based on dehydrated and, consequently, denatured samples. Nonetheless, the presence of PEG had a discernible effect on the structure of the hydrogels, as evidenced by the comparison between AG-3X (dense) and AG-3X* (porous) formulations, which revealed the presence of open and interconnected pores.

At the macrostructure level, the resulting porosity was evenly distributed irrespective of the hydrogel composition, as shown in Figure 2b. The 3D reconstruction clearly demonstrated that the 3D environment was profoundly altered by the addition of PEG, as depicted in Figure 2c. The pores formed exhibited consistent surface characteristics across all hydrogels, in contrast to non-emulsified hydrogels, which contained only a few pores due to entrapped air bubbles.

Regardless of the formulation differences, the overall porosity remained consistent when PEG was introduced and emulsified, as shown in Figure 2d. The hydrogel fraction was approximately 80%, which contrasted with the dense hydrogel control (100%). Morphological analysis of the average pore area, as presented in Figure 2e, indicated a similar pore surface area for all hydrogel compositions, albeit with varying data dispersions.

In summary, the liquid–liquid emulsion method effectively induced porosity in the AG hydrogels. This porosity does not depend on the AG hydrogel concentration, which included AG-1X* (1% *w*/*v* alginate, 2% *w*/*v* gelatin), AG-3X* (3% *w*/*v* alginate, 6% *w*/*v* gelatin), and AG-5X* (5% *w*/*v* alginate, 10% *w*/*v* gelatin), both in terms of volumetric fraction and area.

### 3.2. Spheroids Formation and Inclusion in the Hydrogel

Following a 24 h aggregation step, the inclusion of hiPSCs spheroids (around 1.10^5^ units, with a 145 ± 19 µm diameter for the AG08C5 and 143 ± 22 µm diameter for the SCTi003) in AG-nX* formulations was achieved by gentle homogenization using a pipette under sterile conditions (as shown in Figure 3a). A similar embedding could be performed for dense hydrogel.

The post-inclusion viability analysis after cross-linking revealed a strong calcein-positive signal (green) within the spheroids, and no necrosis signal (red) was observed at the periphery of the spheroids (Figure 3b). Only the centers of the spheroids exhibited a few dead cells. This suggests that the spheroid inclusion method and the mechanical shear generated during embedding preserved the integrity and viability of the spheroids. Validation of this stage marks the start of the overall experiment, which is shown in Appendix A.

### 3.3. Impact of Hydrogel Porosity and Mechanical Properties on Spheroids Growth

Spheroids of AG08C5 included in the dense formulations displayed protuberances and a preference for growth along a specific axis after 7 days of culture. This growth behavior was consistent across hydrogels of varying alginate-gelatin concentrations and, therefore, different mechanical properties (Figure 4a). In contrast, porous formulations exhibited variable growth and spreading profiles for different hydrogel compositions Figure 4b).

Furthermore, spheroid growth appeared to be directly correlated with the increased concentrations of alginate and gelatin in the porous hydrogels. In fact, their morphology exhibited more spreading in AG-5X* than in AG-3X*, and more spreading in AG-3X than in AG-1X*. Given that porosity is similar for all AG-nX hydrogel formulations, the variation in viscoelastic properties could be the primary factor significantly influencing spheroid growth.

To study spheroids growth from two independent hiPSCs lines, monitoring experiments were carried out and quantified through optical microscope acquisitions (Figure 5).

A comparison of the area occupied by spheroids before inclusion and after 14 days of maturation in gel one reveals a growth phenomenon. After maturation, the surface area occupied appears to be directly related to the AG-nX* (n = 1, 3, 5) formulation used, as shown in Figure 5a,b, regardless of the cell line studied.

In particular, the role of the hydrogels compositions was reflected in the growth profile of both AG08C5 and SCTi003 spheroids after maturation. From a similar diameter upon inclusion in the hydrogels (area of 13.4 × 10^5^ µm^2^, expressed in log10 in Figure 5c), spheroid growth after 14 days of culture, in AG-5X was significantly higher than in AG-1X (respectively, up to 93.5 × 10^5^ and 22.3 × 10^5^ mm^2^). On average, the mean surface area of spheroids was increased up to 7-fold in AG-5X, while it did not double in AG-1X, with AG-3X showing intermediate results. Notably, the use of different cell lines did not show statistically significant differences in growth (Figure 5d).

These results demonstrate that in the presence of isoporosity, the growth of induced stem cell spheroids is directly proportional to the increase in the hydrogels’ viscoelastic properties, regardless of hiPSCs origin.

To confirm that the observed effect on spheroid growth was primarily due to viscoelasticity and not influenced by other factors, the different hydrogel formulations were deposited side-by-side within a single cultured sample (as illustrated in Figure 6a). In both cases, whether with AG08C5 (Figure 6b) or SCTi003 (Figure 6c), the area occupied by spheroids at D14 remained dependent on the AG-nX* (n = 1, 3, 5) formulations, similar to what was observed in isolated hiPSCs-laden hydrogels (Figure 5a,b). This suggests that the observed phenomenon was not related to any unforeseen diffusion or biochemical-related effect in the hydrogels. On the contrary, these results emphasize that the growth phenomenon dependent on viscoelastic properties is consistently observed even when different hydrogel compositions are in close contact.

## 4. Discussion

In this study, we aimed to investigate the influence of both porosity and viscoelastic properties on the growth of human induced pluripotent stem cells (hiPSCs) in the absence of growth factors.

Alginate–gelatin hydrogels, known for their affordability, high biocompatibility, and tunable viscoelastic properties, were employed as the base material for this study [42,46,50,51]. Further to this, viscoelastic measurements using Dynamic Mechanical Analysis (DMA) were conducted on solid gels to assess material characteristics in direct contact with the cells. By altering the concentration, a significant variation in viscoelastic properties was achieved within the AG-nX hydrogel (n = 1, 3, 5). Comparable studies on stem cells have traditionally been confined to a narrower range, spanning mostly from Pa [52], a few kPa [53,54], to tens of kPa [27,55]. Additionally, AG-nX hydrogels display pronounced variations in viscosity, as evidenced by a decrease in relaxation time proportional to hydrogel concentration. Recent studies indicate that the viscous properties of the environment promote spheroids growth [44,46,56]. It is worth noting that the cross-linking method remained consistent across all our conditions (Ca^2+^ chelation and transglutaminase action at 37 °C for 10 min), ensuring that the cells were subjected to a similar culture protocol regardless of the hydrogel’s varying viscoelastic properties. The findings derived from our investigation illustrate that important variations of the elastic properties (E) and relaxation time (τ_1/2_) foster the growth of spheroids. Nevertheless, we are unable to isolate the individual effects of elasticity or viscosity. Hence, we focused on elucidating the comprehensive viscoelastic impact. The alginate–gelatin tandem used has been extensively described in terms of their excellent versatility [42,46,51] and consequently, it presents an ideal resource for investigating spheroidal growth within this milieu, predicated exclusively upon disparities in viscoelastic properties among AG-nX formulations.

In terms of cell–material interaction, the primary adhesion protein available for the cells to interact with the hydrogel is the RGD pattern provided by the gelatin [57]. Despite variations in the RGD adhesion pattern concentration (proportional to the gelatin concentration, hence greater in porous AG-5X), Figure 4 demonstrated that, in the absence of porosity and irrespective of gelatin concentration, spheroid growth remained significantly limited. The variation in the RGD pattern is therefore unlikely to be the cause of the observed differences in growth. It is suggested that the restricted growth observed in dense gels may be attributed to the fact that, in such conditions, cells are required to degrade the alginate–gelatin cross-linked matrix to occupy the available space. This notion is supported by the protuberances observed on the spheroids (Figure 4a), which exhibit polarized growth within a limited volume. This underscores the significance of porosity as a critical factor in the growth of stem cell spheroids embedded in a dense matrix like alginate–gelatin.

We hypothesized that the presence of porosity offers a local volume conducive to spheroid growth without necessitating the degradation or remodeling of the matrix by the cells. In this study, AG-nX hydrogels (n = 1, 3, 5) were rendered porous using a water-in-water emulsion approach, which is a unique method for generating porosity within a hydrogel while including a cellular filler. Most porosity induction methods are typically applied upstream of cellularization [58,59] or involve processes such as bioprinting to create microporous scaffolds [60]. Moreover, the presence of porosity in a hydrogel (here not significantly different between formulations) is recognized as a critical factor, particularly for the growth and proliferation of stem cell [25,52,61,62,63].

In the presence of similar porosities, viscoelastic properties appear to significantly impact the ability of spheroids to grow and increase in volume within the different hydrogels. While a visual trend toward the positive impact of material elasticity on spheroid growth was observed in both cell lines, a significant difference between AG-1X* and AG-5X* at D14 was statistically demonstrated. This is also in line with what has been established about the viscoelastic properties of the 3D environment, since the role of viscoelastic properties and relaxation time have been shown to have a direct impact on stem cell migration, proliferation, and differentiation [27,53,64,65,66,67,68,69,70,71]. Material elasticity, notably represented by Young’s Modulus, impacts stem cell physiology at various levels [26,71,72,73], but this has been studied less extensively in a 3D environment. Hydrogel relaxation time, another fundamental viscoelastic parameter, is less well-described but is known to promote cell–material interaction, leading to cell spreading and proliferation [48,56,74]. The use of hiPSCs models (in our case the AG08C5 line and the SCTi003 commercial line, whose pluripotency has been characterized and verified), allows extrapolating the results observed to all artificial stem cells, given the similarity in the biological effects observed.

The protocol presented in this study complements several studies based on similar methodologies, but examining the fate of mesenchymal stem cell spheroids [75,76,77] or cancer cell spheroids [78,79,80] in a porous hydrogel. Most similar studies are carried out in non-porous hydrogels and involve non-aggregated stem cells [5,27,54,81,82]. Also, given that differentiation protocols for hiPSCs can extend up to 21 days [83], it would be relevant to extend the duration of the study to evaluate the long-term impact of environmental mechanical properties on spheroid growth. While the recent emergence of studies involving spheroids, microspheroids or cell aggregates, encapsulated in porous or non-porous hydrogels, is evident, there is no consensus on a standardized method thus far.

The unique aspect of the approach presented here is its focus on induced stem cell spheroids evolving in different hydrogels with the same biochemical composition and macrostructure but with a significant range of viscoelastic variations. This investigation is the first to demonstrate that a mechanically modulated microenvironment induces diverse spheroid behavior without the influence of other factors [40]. Therefore, it has the potential to offer a complementary approach to current techniques, with finely controlled macroporosity spanning from the micrometer to the millimeter scale. As well, it allows us to move towards a bioprocess combining different formulations together to create a truly complex object. Future steps in this approach will concentrate on the study of cell differentiation to further elucidate the impact of such an environment on hiPSCs cell fate.

## 5. Conclusions

The alginate–gelatin combination serves as an excellent biomaterial for promoting the growth of induced pluripotent stem cell spheroids within a cost-effective and highly reproducible 3D environment. It is important to note that, in addition to the crucial role of viscoelastic properties in influencing cell growth, the presence of porosity within the hydrogel matrix is also a significant factor. In the absence of specific biochemical growth factors, hiPSCs spheroids can exhibit varying growth rates within a single scaffold composed of hydrogels with different viscoelastic properties.

The method described in this study is robust and versatile, as it demonstrates consistent growth phenomena across different artificial stem cell lines. Therefore, this approach represents a valuable tool for investigating cell growth by manipulating the viscoelastic properties of the environment alone. It offers a platform to further explore the impact of mechanical cues on stem cell behavior and could have important implications in tissue engineering and regenerative medicine.

## Figures and Tables

**Figure 1 bioengineering-10-01418-f001:**
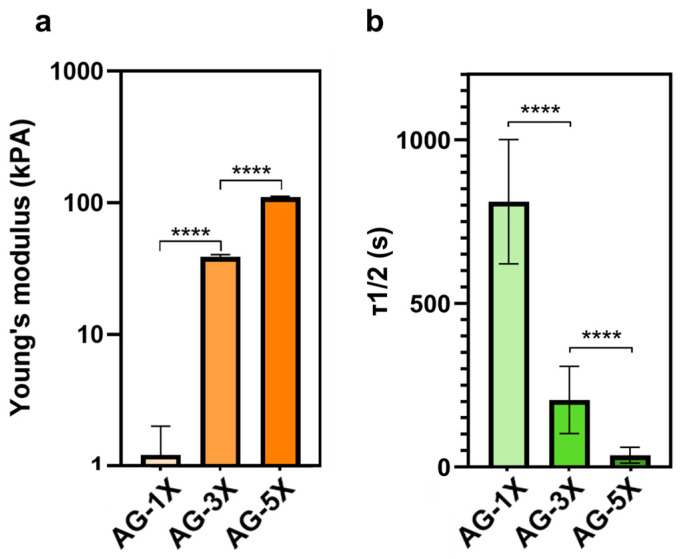
Viscoelastic properties of AG-nX (n = 1, 3, 5) hydrogels at 37 °C. (**a**) Young’s modulus of AG-nX (n = 1, 3, 5) at 37 °C. One-way Anova with Tukey’s multiple comparison test. (**b**) Mid-stress relaxation time at 37 °C. Results presented as mean ± SD. ****: *p* < 0.0001 (at least n = 3 per condition).

**Figure 2 bioengineering-10-01418-f002:**
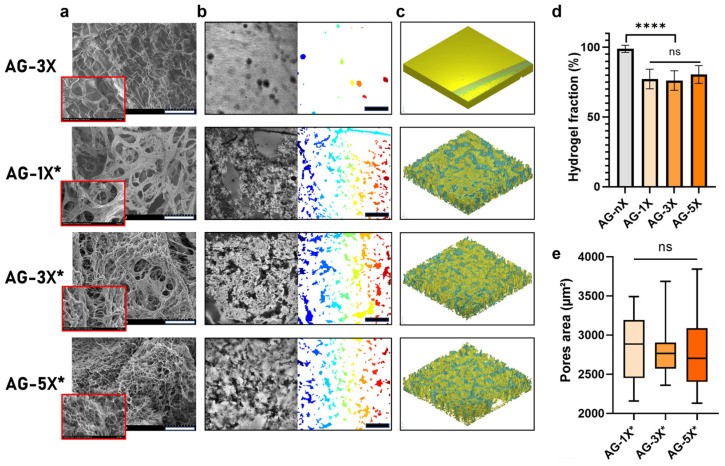
AG-nX (n = 1, 3, 5) hydrogels porosity and surface properties characterization (**a**) SEM surface appearance of AG-nX (10 kV in backscattered electron mode at 300× magnification). Scale bar = 100 µm. (**b**) Quantification of pore surface for each hydrogel with a Matlab^®^ algorithm (* represents porous hydrogels). Scale bar = 300 µm. (**c**) Volume representation of AG-nX porosity (h = 200 µm, l = 1400 µm, L = 1400 µm) (**d**) Fraction of hydrogel on porosity. (* represents porous hydrogels). One-way Anova with Tukey’s multiple comparison test versus AG-3X. (**e**) Average area in AG-nX porous. Kruskal Wallis with Dunn’s multiple comparison test. Results presented as mean ± SD. ns: no significance, ****: *p* < 0.0001 (at least n = 3 per condition).

**Figure 3 bioengineering-10-01418-f003:**
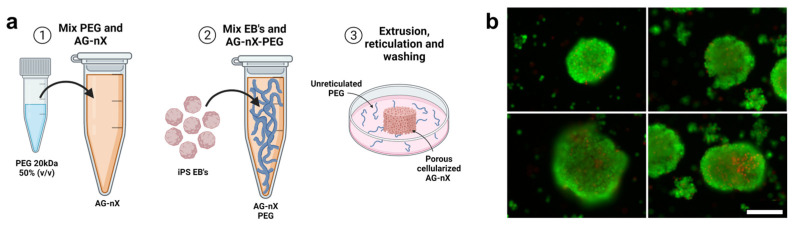
Spheroid inclusion process in hydrogels and post-inclusion viability analysis. (**a**) Preparation of porous hydrogel, inclusion of AG08C5 spheroids, deposition of 50 µL of sample and cross-linking of AG-nX (n = 1, 3, 5) a 37 °C. (**b**) Viability of spheroids post-inclusions (live-dead assay). Scale bar = 200 µm.

**Figure 4 bioengineering-10-01418-f004:**
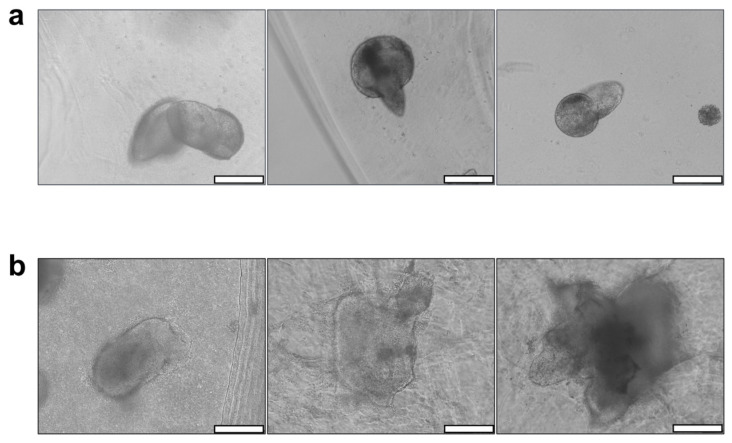
Impact of AG-nX (n = 1, 3, 5) hydrogel porosity on AG08C5 hiPSCs spheroid growth at D7. (**a**) Phenotype of AG08C5 spheroids included in dense AG-nX hydrogels. (**b**) Phenotype of AG08C5 spheroids included in porous hydrogels AG-NX. scale bar = 100 µm.

**Figure 5 bioengineering-10-01418-f005:**
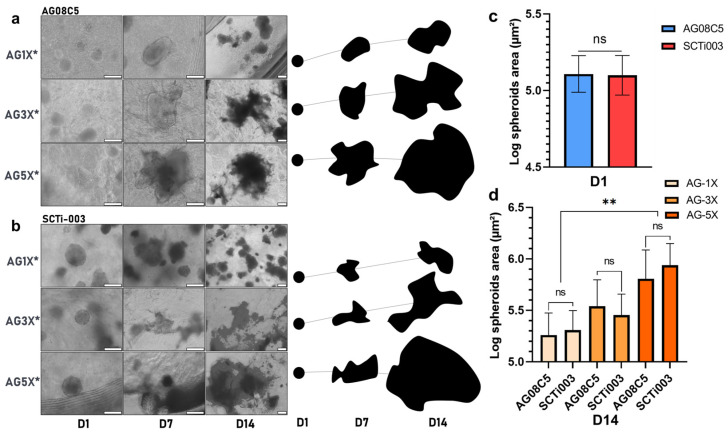
Monitoring and quantification of hiPSCs spheroids growth of in porous AG-nX hydrogels (**a**) AG08C5, representative optical microscope acquisition and representation on growth area at D1, D7 and D14. scale bar = 200 µm. (**b**) SCTi-003, representative optical microscope acquisition and representation on growth area at D1, D7 and D14. scale bar = 200 µm. (**c**) Spheroid area before inclusion in AG-nX*. Mann-Whitney test, *p* = 0.418. (**d**) Comparison of spheroids growth of both cell lines in AG-nX* condition at D14. One-way Anova with Tukey’s multiple comparison test. Results presented as mean ± SD. ns: no significance, **: *p* < 0.01 (at least n = 3 per condition).

**Figure 6 bioengineering-10-01418-f006:**
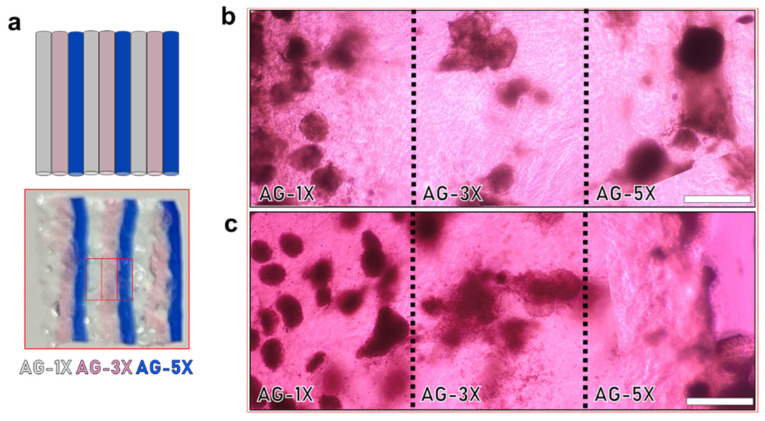
Impact of viscoelastic properties of AG-nX hydrogels (n = 1, 3, 5) on the growth of hiPSCs spheroids in transversal culture at D14. (**a**) Representation of alternating hydrogels in transverse culture (AG-1X, AG-3X and AG-5X, extruded 3 times). (**b**) AG08C5 spheroids deposited in the same well at D14. (**c**) SCTi-003 spheroids deposited in the same well at D14. Scale bar = 500 µm.

## Data Availability

The datasets used and analyzed during the current study are available from the corresponding authors upon reasonable request.

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
