# Peer review of "Human Induced Pluripotent Spheroids’ Growth Is Driven by Viscoelastic Properties and Macrostructure of 3D Hydrogel Environment"

_bioengineering, 2023, doi:10.3390/bioengineering10121418_

Round 1
Reviewer 1 Report
Comments and Suggestions for Authors
This is an interesting research evaluating the effect of several hydrogel micro-environments for the development of HiPSCs. In general the manuscript is well write. However, during my review process several concerns have raised, particularly related with M&M and Results sections:
1. It is necessary to add a figure indicating the study design, showing the workflow of your study.
2. It is necessary to re-evaluate your data by using a GLMM. This kind of statistical approach could improve your results and expand your final conclusions.
3. Please, you should present your results without any discussion. You use references to premature explain your results. You have the discussion section to do this.
4. Please, arrange the section results to avoid " (Error! Reference source not found..
Reviewer 2 Report
Comments and Suggestions for Authors
Manuscript ID: bioengineering-2744477
Title: Human induced pluripotent spheroids growth is driven by viscoelastic
properties and macrostructure of 3D hydrogel environment
Authors: Lucas Lemarié, Tanushri Dargar, Isabelle Grosjean, Vincent Gache, Edwin Joffrey Courtial, Jérôme Sohier *
1. What is the main question addressed by the research?
The research covers the area of utilization of 3D hydrogels incorporated with stem cells for tissue engineering applications.
2. Do you consider the topic original or relevant in the field? Does it
address a specific gap in the field?
The research topic is original and it covers an important point of research.
3. What does it add to the subject area compared with other published
material?
It add the utilization of 3D hydrogel based on biomaterials and biodegradable materials using stem cells driven from human. Moreover, the materials used are more effective than the published research.
4. What specific improvements should the authors consider regarding the
methodology? What further controls should be considered?
The authors should add FT-IR, EDX, BET and xrd analysis of the pure materials to evaluate their performance more specific.
5. Are the conclusions consistent with the evidence and arguments presented
and do they address the main question posed?
The conclusions are clear and enough to address the main issue discussed in the manuscript.
6. Are the references appropriate?
The references need to be updated and to add more references of other researchers, techniques and materials comparing to this work.
7. Please include any additional comments on the tables and figures.
Tables and figures are fine but additionally the authors should add a summarized graphical abstract explaining the work steps.
8. The references covering the electrospinning technique suggested by me are
(a) Preparation of Antibacterial Electrospun PVA / ï¼²egenerated Silk Fibroin Nanofibrous Composite Containing Ciprofloxacin Hydrochloride as a Wound Dressing.
(b) http://dx.doi.org/10.3390/pharmaceutics15051518.
(c) https://doi.org/10.1016/j.carbpol.2021.118952
(d) https://doi.org/10.25972/OPUS-24126
(e) https://doi.org/10.3390/ijms23052662
(f) J. Mater. Chem. B, 2022,10, 1486-1507
(g) https://doi.org/10.3390/bioengineering6040113
(h) https://doi.org/10.1177/2041731417712073
(i) https://doi.org/10.1016/j.ijbiomac.2022.11.213
(j) https://doi.org/10.1021/acsami.2c03705
(k) https://doi.org/10.2147/IJN.S363777
Reviewer 3 Report
Comments and Suggestions for Authors
This paper is a thorough characterization of the impact of hydrogel biomechanics on the growth of induced pluripotent stem cell spheroids. The study is well described and rich in experimental results. Apart from minor reporting issues, commented below, I think this work is a useful contribution to the tissue engineering literature.
1. References to figure parts in the main text should be formatted according to the publisher's template. For example, on line 198, instead of "Figure 1. a." (where Figure is written in Bold) please write simply Figure 1a (without Bold). Proceed similarly for other figures, too.
2. Instead of an Appendix, I would prefer to have a Supplementary Material file because it would offer the chance to give further explanations pertaining the model. Whatever the decision of the authors, Appendix or Supplementary Material, Figure A1 needs to be commented in the main text, panel by panel. A single comment on panel b does not justify its presence. The experimental results from Figure A1a are not exploited at all. Furthermore, Figure A1d is actually a table, so I would separate it from the other panels and comment it accordingly.
3. Minor revisions, listed in the format old_text => proposed_revision
Line 111: The storage and loss moduli (E' and E'', respectively) obtained =>
The measured storage and loss moduli (E' and E'', respectively)
Line 208: Figure 2. A, => Figure 2a
Also, no Bold is needed for "Figure 2".
Line 222: In the caption of Figure 2a, I would add the SEM settings (10kV in backscattered electron mode at 300X magnification) to save time for the reader. Otherwise, she/he needs to go back to section 2.3.
Line 269: gelation => gelatin
Lines 297-299: Please revise the sentence "These results ... isoporosity.". I had a hard time trying to understand it.
Comments on the Quality of English LanguageMinor mistakes of English usage are described in point 3 of my report.
Round 2
Reviewer 1 Report
Comments and Suggestions for Authors
All my concerns were addressed by the authors
Reviewer 2 Report
Comments and Suggestions for Authors
I would like to thank the authors for their efforts. Good luck